# Application of Sonographic Assessments of the Rate of Proximal Progression to Monitor *Protobothrops mucrosquamatus* Bite-Related Local Envenomation: A Prospective Observational Study

**DOI:** 10.3390/tropicalmed8050246

**Published:** 2023-04-24

**Authors:** Feng-Chen Chen, Ahmad Khaldun Ismail, Yan-Chiao Mao, Chih-Hsiung Hsu, Liao-Chun Chiang, Chang-Chih Shih, Yuan-Sheng Tzeng, Chin-Sheng Lin, Shing-Hwa Liu, Cheng-Hsuan Ho

**Affiliations:** 1Department of Emergency Medicine, Kaohsiung Armed Forces General Hospital, Kaohsiung 80284, Taiwan; 2Department of Emergency Medicine, Tri-Service General Hospital, National Defense Medical Center, Taipei 11402, Taiwan; 3Department of Emergency Medicine, Faculty of Medicine, Universiti Kebangsaan Malaysia Medical Centre, Kuala Lumpur 56000, Malaysia; 4Division of Clinical Toxicology, Department of Emergency Medicine, Taichung Veterans General Hospital, Taichung 407219, Taiwan; 5Health Service and Readiness Section, Armed Forces Taoyuan General Hospital, Taoyuan 325, Taiwan; 6National Tsing Hua University, College of Life Sciences, Hsinchu 300044, Taiwan; 7Division of Plastic Surgery, Department of Surgery, Tri-Service General Hospital, National Defense Medical Center, Taipei 11402, Taiwan; 8Division of Cardiology, Department of Internal Medicine, Tri-Service General Hospital, National Defense Medical Center, Taipei 11402, Taiwan; 9Institute of Toxicology, College of Medicine, National Taiwan University, Taipei 10051, Taiwan

**Keywords:** *Protobothrops mucrosquamatus*, rate of proximal progression, freeze-dried hemorrhagic antivenom, point-of-care ultrasound, snakebite

## Abstract

Patients bitten by *Protobothrops mucrosquamatus* typically experience significant pain, substantial swelling, and potentially blister formation. The appropriate dosage and efficacy of FHAV for alleviating local tissue injury remain uncertain. Between 2017 and 2022, 29 snakebite patients were identified as being bitten by *P. mucrosquamatus*. These patients underwent point-of-care ultrasound (POCUS) assessments at hourly intervals to measure the extent of edema and evaluate the rate of proximal progression (RPP, cm/hour). Based on Blaylock’s classification, seven patients (24%) were classified as Group I (minimal), while 22 (76%) were classified as Group II (mild to severe). In comparison to Group I patients, Group II patients received more FHAV (median of 9.5 vials vs. two vials, *p*-value < 0.0001) and experienced longer median complete remission times (10 days vs. 2 days, *p*-value < 0.001). We divided the Group II patients into two subgroups based on their clinical management. Clinicians opted not to administer antivenom treatment to patients in Group IIA if their RPP decelerated. In contrast, for patients in Group IIB, clinicians increased the volume of antivenom in the hope of reducing the severity of swelling or blister formation. Patients in Group IIB received a significantly higher median volume of antivenom (12 vials vs. six vials; *p*-value < 0.001) than those in Group IIA. However, there was no significant difference in outcomes (disposition, wound necrosis, and complete remission times) between subgroups IIA and IIB. Our study found that FHAV does not appear to prevent local tissue injuries, such as swelling progression and blister formation, immediately after administration. When administering FHAV to patients bitten by *P. mucrosquamatus*, the deceleration of RPP may serve as an objective parameter to help clinicians decide whether to withhold FHAV administration.

## 1. Introduction

In Taiwan, there are six major species of medically important venomous snakes, namely, *Protobothrops mucrosquamatus* (Günther, 1864), *Trimeresurus stejnegeri* (Schmidt, 1925), and *Deinagkistrodon acutus* (Günther, 1888) in the Crotalinae subfamily [family: Viperidae]; *Daboia siamensis* (Smith, 1917) in the Viperinae subfamily [family: Viperidae] and *Bungarus multicinctus* (Blyth, 1861) and *Naja atra* (Wüster, 1991) in the Elapidae family [1,2]. *P. mucrosquamatus*, also called the Taiwan habu, is one of the most encountered medically important snakes [1,3]. Patients bitten by *P. mucrosquamatus* typically experience painful, substantial tissue swelling, and local cyanosis. Coagulopathy and thrombocytopenia are uncommon with *P. mucrosquamatus* bites [4]. The toxin profiles of *P. mucrosquamatus* include snake venom metalloproteinase (SVMP; 29.4%), C-type lectin (CLEC; 21.1%), snake venom serine protease (SVSP; 17.6%) and phospholipase A_2_ (PLA_2_; 15.9%) [5,6]. SVMP and PLA_2_ are responsible for capillary wall leakage and muscle damage and resulting in local cyanosis and substantial tissue swelling [7,8].

In Taiwan, there are four types of horse-derived antivenom, which are derived from the lyophilized form, ammonium sulfate-precipitated F(ab’)_2_ fragment, manufactured by the National Health Research Institutes, Miaoli, and distributed by the Centers for Disease Control [9]. Both antivenoms are bivalent; one is used against *P. mucrosquamatus* and *T. stejnegeri* venom, also called freeze-dried hemorrhagic antivenom (FHAV), and the other is used to treat *N. atra-* and *B. multicinctus*-bitten patients, named freeze-dried neurologic antivenom (FNAV) [9]. The Taiwan National Poison Control Center (PCC) recommends 2–4 vials for patients bitten by *P. mucrosquamatus* [9] based on the neutralizing ability of antivenom against the median lethal dose (LD_50_) of envenomation in animal studies [10].

Antivenom is the antidote for snakebites and is suggested to be administered as soon as possible. Even under the administration of FHAV, patients bitten by *P. mucrosquamatus* usually developed progressive swelling, and clinicians had difficulties realizing what volume of administered antivenom was sufficient. Therefore, an objective clinical measurement of the progress after each dose of antivenom is necessary for the clinician to avoid the unnecessary administration of a large volume of antivenom, which results in high medical costs and possible serum sickness. The goal of this study is to propose the rate of proximal progression (RPP) in the local envenomated tissue as an objective parameter to evaluate when to withhold FHAV. Furthermore, we discuss the effectiveness of FHAV in improving local tissue injury, which is questionable.

## 2. Materials and Methods

### 2.1. Patient Recruitment and Characteristics

This was a prospective observational study of snake-bite envenomation cases from January 2017 to September 2022. We recruited snakebite patients, including definite, suspected, and clinical diagnoses. Definite cases indicate that the patient can provide a snake body, pictures, or toxin analysis from the serum or bullae fluid. Suspected cases indicate that the patient can identify the characteristics of the snake. Clinical diagnosis cases mean that the patient cannot provide evidence of snakes, and clinicians diagnose based on the clinical scenario after excluding other injuries [4,11]. Approval was obtained from the Institutional Review Board (IRB) of the Tri-Service General Hospital (IRB No.: 2-107-05-039). Each patient provided written informed consent prior to enrollment. The following data were extracted: sex, age, emergency department length of stay (ED-LOS), bite site, total doses of antivenom, worst tissue injury, disposition, laboratory findings, sonographic findings, and possible sequelae. The worst tissue swelling level was classified with Blaylock’s categories [12]. The Blaylock swelling level was categorized as (1) minimal (local swelling at the bite site), (2) mild (swelling involving the whole hand or foot), (3) moderate (swelling from foot to the proximal thigh or from hand to shoulder), (4) severe (swelling from foot to the groin or from the hand to the ipsilateral chest wall) or (5) gross (swelling from the foot to the trunk or from hand to contralateral chest wall, abdomen, and neck) [12]. In all recruited snakebite patients, we further analyzed the definite *P. mucrosquamatus*-bitten patients who were divided into 2 groups based on the severity of the local tissue injury. The severity of swelling was categorized as minor in Group I and as mild to severe in Group II. All patients bitten by *P. mucrosquamatus* received the point-of-care ultrasound (POCUS) protocol, including measurement of the rate of proximal progression (RPP) [13,14]. Clinicians on duty had the discretion to administer the antivenom guided by the RPP, or not. Afterward, we retrospectively grouped the patients. Those who received treatment according to RPP guidelines were placed in group IIA, while those who were given more antivenom to reduce local tissue injury were placed in group IIB. After discharge from the hospital, all patients were followed up by telephone or outpatient department (OPD) for serum sickness symptoms and local tissue complete remission until the local injury improved. Remission of local injury was defined as the patient feeling that the swelling improved and neither pain by themselves.

### 2.2. Patient Management: Antivenom Administration and POCUS Monitoring

The type, administration frequency, and volume of antivenom were determined by the clinician. The antivenom was given in doses of 1 or 2 vials at a time and administered every 1.5–2 h. The frequency and total amount of antivenom given depended on the clinician’s judgment. The FHAV and FNAV were manufactured by the National Health Research Institutes, Miaoli, and distributed by the Centers for Disease Control [9], and each vail of antivenom was filled with more than 1000 Tanaka units, which could neutralize 1 minimum venom lethal dose (MLD) [15,16]. The clinicians used the FHAV for patients bitten by *P. mucrosquamatus* and *T. stejnegeri* and FNAV for patients bitten by *N. atra* and *B. multicinctus* and observed the patients without envenomation if the snake was identified as Colubridae or nonidentified. All definite *P. mucrosquamatus*-bitten patients were examined using the POCUS protocol [13] to evaluate wound progression. We used CX50 sonography machines (Philips ultrasound, INC. Software Version: 4.0.2; 22100 Bothell Everett Highway, Bothell, WA, USA), which were equipped with 15-MHz linear probes. In this protocol, we used 15-MHz linear probes to measure the location of swelling and the diastolic retrograde arterial flow (DRAF) of the compressed artery [13]. The clear borderline between the cobblestone sign and the normal tissue was marked on the patient’s skin, and we evaluated the distance of progression over the following hours until the progression stopped. The distance between the 2 markers divided by the number of hours was defined as the rate of proximal progression, RPP (cm/hour) [14].

### 2.3. Toxin analysis: Enzyme-Linked Immunosorbent Assay (ELISA)

The 96-well half-area plates were coated overnight with the control serum sample (10 μg/well), the patient’s serum or a bullae fluid sample (10 μg/well) and bovine serum albumin (BSA; 10 μg/well) and were blocked with 5% skimmed milk dissolved in phosphate-buffered saline (PBS). The primary antibody was the specific anti-*P. mucrosquamatus* yolk polyclonal immunoglobulin Y (IgY) antibody [17,18], purified from immunized chickens, was used at a 5000-fold dilution and incubated for 1 h at 37 °C. After routine washing, the secondary antibody, which was HRP-conjugated donkey anti-chicken IgY (Jackson ImmunoResearch, West Grove, PA, USA), was applied at a 10,000-fold dilution for 1 h at 37 °C. After the final wash, the binding activities were assessed with 3,3′,5,5′-tetramethylbenzidine (TMB) substrate (BD Biosciences, San Jose, CA, USA), the reaction was stopped with 1 N HCl, and the optical density at 450 nm (OD_450nm_) was measured using a Synergy HT ELISA reader (Bio-Tek Instruments, Winooski, VT, USA). The ELISA results were obtained in triplicate wells for each sample.

### 2.4. Statistical Analysis

We report the data as median, range, and percentage. For continuous variables, we compared these groups using the *Mann*–*Whitney U* test and the *Kruskal*–*Wallis* test. For categorical variables, we used the chi-squared test. All statistical tests were 2-sided, and *p*-values less than 0.05 were considered statistically significant. We conducted statistical analyses for the clinical study using IBM SPSS Statistics version 22 (IBM^®^ SPSS^®^ Statistics 22, Chicago, IL, USA).

## 3. Results

### 3.1. Characteristics of the Recruited Patients

From January 2017 to September 2022, 51 patients were recruited. The kinds of snakes were classified as Viperids, Elapids, Colubridae, and nonidentified. Thirty-nine patients (76%) were bitten by Viperids, three patients (6%) by Elapids, and seven patients (14%) by Colubridae. Two nonidentified patients (4%) claimed that they were bitten by snakes without envenomated symptoms during the observation period (6.75 ± 1.77 h), and we suspected that the snakes were nonvenomous or that these were dry bites by venomous snakes. The Viperids in this study included *P. mucrosquamatus* and *T. stejnegeri*, while the Elapids included *B. multicinctus* and *N. atra*. Patients bitten by the Colubridae family, which includes *Boiga kraepelini* (two patients), *Cyclophiops major* (two patients), *Fowlea flavipunctatus*, *Elaphe carinata*, and *Lycodon rufozonatus*. Among all types of snakebites, males accounted for 29 patients (74%) in Viperids, three patients (100%) in Elapids, three patients (43%) in Colubridae, and one patient (50%) in nonidentified (*p*-value = 0.21). The median age of snakebite patients was 54.00 years (range 5–82 years) in Viperids, 51.00 years (range 5–59 years) in Elapids, 39.00 years (range 27–70 years) in Colubridae, and 32.50 years (28 and 37 years) in nonidentified (*p*-value = 0.44). The average age of snakebite patients was 50.97 ± 19.57 years in Viperids, 38.33 ± 29.14 years in Elapids, 45.86 ± 16.73 years in Colubridae, and 32.5 ± 6.36 years in nonidentified. June to November was the most frequently encountered period every year (69% in Viperids, 67% in Elapids, 71% in Colubridae; *p*-value = 0.96). Viperid bite patients received a higher volume of antivenom [median 6.00 vials, range 1.0–22.0 vials] than other types of snakes [Elapid median 4.00 vials, Colubridae median 1.00 vial] (*p*-value < 0.0001). Viperid bite patients also had a longer ED-LOS [median 20.00 h, range 1.5–64 h] than other types of snakes [Elapid 4.50 h, Colubridae 6.00 h, nonidentified 6.75 h] (*p*-value < 0.001). Most patients (45 patients, 88.2%) could be discharged from the ED, and there was no difference in the disposition among different types of snakes (*p*-value = 0.13).

### 3.2. Characteristics of the Patients Bitten by P. mucrosquamatus

Furthermore, in the Viperid-bitten patients, we analyzed the definite *P. mucrosquamatus*-bitten patients who provided the snake body (Figure 1), pictures, or toxin analysis. There was a total of 29 patients, and we divided them into two groups based on the different progression of local envenomation (Table 1). We used Blaylock’s categories [12] to classify the worst swelling status of each patient as minimal, mild, moderate, severe and gross. None developed as severe as “gross” in these patients. Group I had seven patients (24%) who had been categorized as minimal, and Group II included the other twenty-one patients (76%) who had been labeled mild to severe (*p*-value < 0.0001). There was no gender difference between the two groups (*p*-value = 0.06). The median age of the patients was 59.00 years (range 25–65 years) in Group I and 57.50 years (range 5–82 years) in Group II (*p*-value = 0.65). The method of identification included the snake body (29% vs. 32%), photos (71% vs. 55%), and toxin analysis (0% vs. 14%; *p*-value = 0.54). The encountered seasons were winter (December to February; 0% vs. 9%), spring (March to May; 29% vs. 14%), summer (June to August; 14% vs. 27%) and fall (September to November; 57% vs. 50%; *p*-value = 0.63). Every year, September to November were the most frequently encountered months for patients bitten by *P. mucrosquamatus*. The patients were bitten over the upper limbs [finger, hand, and forearm] (57% vs. 28%) and the lower limbs [toe, foot, ankle, and lower leg] (43% vs. 72%; *p*-value = 0.44). In the clinical presentation, none presented or developed a fever. The Group I patients developed local cyanosis, and the Group II patients had local cyanosis (100%), progressed swelling (100%), wound necrosis (two patients, 3%) and blister/bullae formation (two patients, 3%). None of the Group I patients developed leukocytosis (WBC > 11,000 cells/μL) [normal range: 4500–11,000 cells/μL], and six patients (27%) of the Group II patients had leukocytosis (*p*-value = 0.12). None of the Group I patients developed thrombocytopenia (platelet < 150,000 cells/μL) [normal range: 150,000–400,000 cells/μL], and two patients (9%) of the Group II patients had thrombocytopenia (*p*-value = 0.41). Not all patients had their CK measured, and there was no patient with a high serum CK level (CK > 1000 U/L) [normal range: 39–308 U/L] in Group I and two patients (9%) in Group II (*p*-value = 0.44). On the renal function presented with Kindo grade [19], there was no significant difference, as grade 1 (57% vs. 45%), grade 2 (43% vs. 32%), and grade 3 (0% vs. 23%; *p*-value = 0.38). The Group I patients received a median of 2.00 vials of antivenom (range 1–4 vials), while the Group II patients received a median of 9.50 vials (range 4–22 vials; *p*-value < 0.0001). In addition, the Group I patients had a median emergency department (ED) length of stay (LOS) of 8.00 h (range 3–18 h), and the Group II patients had an ED LOS of 24.50 h (range 13–61 h; *p*-value < 0.0001). Under POCUS monitoring [13], none developed DRAF in the compressed artery in these two groups of patients. All Group I patients were discharged from the ED, and four patients (18%) in Group II were admitted (*p*-value = 0.22). In the telephone or OPD follow-up, none of the patients developed serum sickness. The median days of local tissue injury complete remission were significantly different between Groups I and II (2.00 days vs. 10.00 days; *p*-value < 0.001).

### 3.3. Different Types of Clinical Management Based on the RPP

The clinicians on duty had the discretion to administer antivenom in volumes according to RPP guidelines (Figure 2) or not. We retrospectively grouped patients into two categories: Group IIA, which comprised patients whose antivenom administration was withheld when RPP decelerated, and Group IIB, which comprised patients who received additional volumes of antivenom to reduce local tissue injury even when their RPP decelerated (Figure 3). There were eleven patients in each group, and there were no significant differences in severe swelling condition (*p*-value = 0.59), sex (*p*-value = 0.08), age (*p*-value = 0.21), snake identification method (*p*-value = 0.67), season encountered (*p*-value = 0.12) or bite site (*p*-value = 0.34). The severity of swelling between these two subgroups was mild (82% vs. 73%), moderate (each 18%) and severe (0% vs. 9%; *p*-value = 0.59). The fall (September to November) was the most common season (45% and 55%). The lower limbs (toe, foot, and lower leg) were the most bitten sites (82% and 64%). The patients had leukocytosis (45% vs. 9%; *p*-value = 0.06), thrombocytopenia (18% vs. 0%; *p*-value = 0.14), higher serum CK level (each 9%; *p*-value = 0.73), Kindo grade 1 (27% vs. 64%), grade 2 (36% vs. 27%), and grade 3 (36% vs. 9%; *p*-value = 0.17). The Group IIA patients received a lower volume of antivenom, at 6.0 vials (range 4–10 vials), than the Group IIB patients, who received 12.00 vials (range 8–22 vials; *p*-value < 0.001). Additionally, Group IIA had a shorter ED-LOS of 21.00 h compared to Group IIB’s ED-LOS of 27.50 h (*p*-value = 0.03). There was no difference in the disposition between these two subgroups, including discharge from the ED (73% vs. 91%) and admission (27% vs. 9%; *p*-value = 0.33). Neither of these two groups developed acute compartment syndrome or acute serum sickness during follow-up. There was no difference in complete remission days between Groups IIA and IIB (7.00 days vs. 12.00 days, *p*-value = 0.06). We further examined the subgroup consisting of the majority of patients whose Blaylock’s categories were classified as mild. There were nine patients in Group IIA and 8 patients in Group IIB. In this sub-analysis, Group IIA received a lower volume of FHAV than Group IIB (a median of six vials vs. 11 vials, with a *p*-value of <0.05). There were no significant differences between the genders (78% male vs. 50% male), ED-LOS (a median of 22 h vs. 26.5 h), or complete remission days (a median of 6 days vs. 12 days).

### 3.4. Surgery for Patients Bitten by P. mucrosquamatus

In this study, none of these patients received fasciotomy or fasciectomy for the development of acute compartment syndrome, but two of them received debridement for wound necrosis. Patient 1 is a 55-year-old male in Group IIA who was bitten over the right thumb. He received six vials of antivenom, and the swelling stopped on his forearm. One day after the bite, a hemorrhagic blister developed (Figure 4A,B). The clinician suggested that debridement not increase the volume of FHAV. He was discharged 4 days after admission. Patient 2 is a 71-year-old female in Group IIB. She was bitten on her left foot, and her tissue swelling stopped after six vials of antivenom. Blisters developed 11 h after the snakebite (Figure 4C,D). Unlike the patients in Group IIA, she received another six vials of antivenom in the hope of improving the blister, but this was in vain. Eventually, she received debridement 1 day after the snake bite and was discharged 3 days after admission.

## 4. Discussion

Here, we described the characteristics of snakebite patients (51 patients) in our hospital from January 2017 to September 2022, and viperid was the most frequently encountered. We further analyzed the characteristics of the 29 patients who had been bitten by *P. mucrosquamatus*. These patients were divided into two groups based on Blaylock’s local tissue swelling category: minimal in Group I and mild to severe in Group II. Group II was further divided into IIA and IIB according to the clinical management for the deceleration of RPP. According to the guidance of the RPP deceleration, Group IIA patients received less antivenom (*p*-value < 0.001) but experienced no more complications or sequelae than Group IIB patients who received more volumes of FHAV, ignoring the RPP deceleration. Therefore, when administering FHAV to a patient, the RPP deceleration may serve as an objective parameter to help clinicians decide whether to withhold FHAV administration.

Progressive swelling or blister formation is the major driving force for the clinician to add doses of antivenom in Group IIB. More antivenom causes higher medical costs and possibly a higher incidence of serum sickness [20,21]. Therefore, an objective clinical measurement of the progress after each dose of antivenom is necessary for the clinician to avoid unnecessary administration of a large volume of antivenom. RPP seems to be an effective guide for the clinician when administering FHAV. The application of sonography to determine the margin of the edema and measure the rate of proximal progression at intervals has been widely used in Malaysia and Taiwan as adjuncts to gauge the severity of snakebites and the indication for antivenom therapy [13,14]. Based on the sonography cobblestone sign, the clinician can easily visualize the location and progression of interstitial edema beyond assessments of the gross cyanosis pattern. In this study, we found that when RPP starts to decelerate, administering a greater volume of antivenom did not show additional benefit.

Comparing Groups IIA and IIB, a larger subsequent dose of FHAV was not helpful for Group IIB patients to improve the symptoms of *P. mucrosquamatus* bites, including the prevalence of leukocytosis, thrombocytopenia, high CK level or acute kidney injury, tissue swelling and blister formation, wound necrosis, and disposition. In this study, the timing of complete remission of local tissue injury was associated with the severity of local tissue swelling, not the volume of antivenom. Therefore, tissue remission may still depend on cell regeneration, not the contribution of FHAV. We reasonably questioned the effectiveness of FHAV for local tissue injury. The FHAV was harvested from the horses through immunization by the crude venom of *P. mucrosquamatus* and *T. stejnegeri* and then quantified by the neutralization of the minimal lethal dose and presented as the Tanaka unit [9]. The recommended volumes of FHAV for envenomed patients are estimated from the LD_50_ in the animal model [6,7]. In short, the design and recommended volume of FHAV were focused on reducing the systemic mortality effects of snake venom. In Taiwan, antivenom indeed decreases snakebite-related mortality from 1.4% to 0.04% [22,23], and the serum level of *P. mucrosquamatus* toxin was objectively decreased after FHAV administration [24]. FHAV is effective for the *P. mucrosquamatus*-related systemic effect. However, *P. mucrosquamatus* bite resulted in many local tissue injuries, such as progressed edema, cyanosis, blister formation and wound necrosis [4,11,25,26]. Lin et al. demonstrated the feasibility of using FHAV to reduce swelling in affected limbs [24]. However, our study reveals that patients bitten by *P. mucrosquamatus* who received higher volumes of FHAV, even in the early stages, did not immediately improve local tissue injury and did not necessarily experience shorter complete remission days. In addition to being ineffective for advanced swelling, a greater volume of FHAV may also be unable to reduce or reverse the formation of blisters or necrosis. In this study, the surgical rate for patients bitten by *P. mucrosquamatus* was 7% (two patients), with debridement for wound necrosis as the surgical indication. The risk factors for *P. mucrosquamatus* bit-related wound necrosis are finger as the bitten site and bullae or blister formation [4]. If patients bitten by *P. mucrosquamatus* develop blister formation or wound necrosis, debridement may be superior to increasing the volume of FHAV. To summarize, although antivenoms have been found to be effective in treating systemic envenoming resulting from snakebites, their effectiveness may be limited when it comes to treating the local tissue injuries [25,26,27,28].

We hypothesized that FHAV might neutralize the lethal component of venom but not be effective for the component leading to the progression of local tissue injury. The affinity of FHAV to the different components of *P. mucrosquamatus* still needs further evaluation. For example, FNAV is another antivenom harvested from hoses by the crude venom of *B. multicinctus* and *N. atra* [9]. FNAV has a higher neutralizing ability for SVMP than cytotoxin in the crude venom of *N. atra* [29], but cytotoxin plays a major role in *N. atra*-related dermonecrosis [26,28]. The World Health Organization (WHO) suggests that other methods are needed to evaluate the neutralizing ability of antivenom for venom-related local cytotoxic effects, perhaps through assessments of the minimum hemorrhagic dose (MHD) and minimum myotoxic dose (MMD) [30], not only from the LD_50_. In our previous study, we found that FNAV did not help stop dermonecrosis, as indicated by the MND_50_ assay results [26]. In summary, we doubt the effectiveness of FHAV for the local cytotoxic effect, which may be the reason that Group IIB patients did not have shorter complete remission days or better outcomes even when administered more antivenoms. Further evaluation is needed for the neutralizing test of venom components and local cytotoxic assay in FHAV and *P. mucrosquamatus* venom.

In addition to the questionable effectiveness of FHAV for local envenomation, serum sickness is another concern. Serum sickness is characterized by fever, unexplained rash, arthralgia, and pruritus [21]. Serum sickness is a type III or immune complex-mediated hypersensitivity disease that occurs 4–14 days after exposure to foreign proteins [21,31,32]. There is no definite treatment for serum sickness; therefore, a method for monitoring the effectiveness of antivenom and avoiding overshooting the amount of antivenom is important for clinicians.

## 5. Limitations

There were three limitations to this study. First, when RPP decelerated, the decision about whether to administer more antivenom or withhold it was based on the clinician’s judgment rather than randomized control. Second, due to the small number of patients, we could only establish that when RPP decreased, it was the appropriate time to withhold antivenom. We could not establish a correlation between the absolute value of RPP and the total volume of antivenom required by the patients. Third, complete remission was evaluated subjectively by each patient and lacked objective evaluation. In future studies, objective measures should be used to evaluate the patients’ post-bite function.

## 6. Conclusions

Patients bitten by *P. mucrosquamatus* easily present with progressive tissue swelling even under the administration of FHAV. Administering a higher volume of FHAV may not provide additional benefits when sonographic assessments of RPP deceleration are observed. FHAV does not appear to prevent local tissue injuries, such as swelling progression and blister formation, immediately. When administering FHAV to patients bitten by *P. mucrosquamatus*, the RPP deceleration may serve as an objective parameter to help clinicians decide whether to withhold FHAV administration.

## Figures and Tables

**Figure 1 tropicalmed-08-00246-f001:**
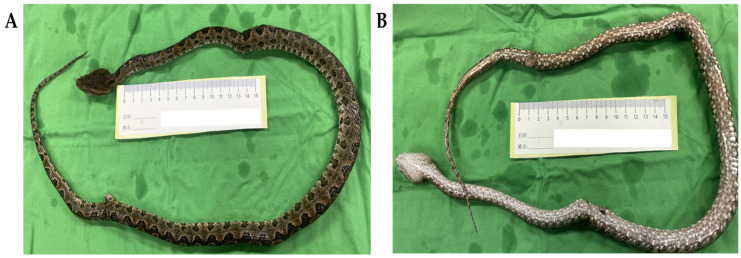
The snake bodies were provided by the patients and identified as *P. mucrosquamatus*. The dorsal view (**A**) and ventral view (**B**) of a snake provided by an 81-year-old male in Group IIA in 2022. The snake’s body length was approximately 45 cm.

**Figure 2 tropicalmed-08-00246-f002:**
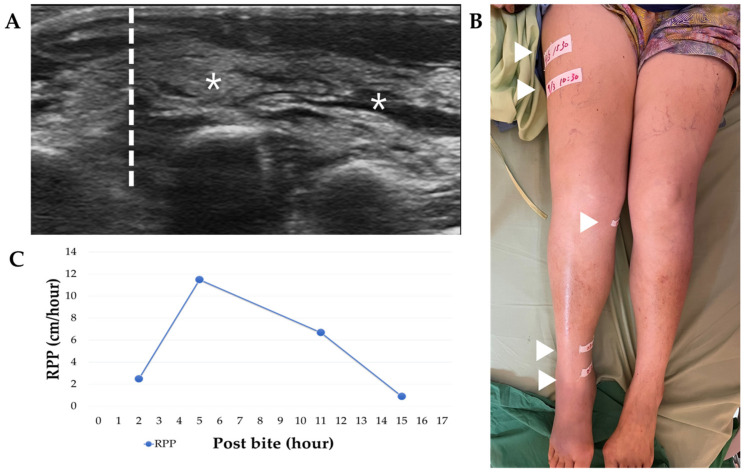
The patient is a 60-year-old female in Group IIA in the moderate Blaylock’s category. She was bitten on the right dorsal foot, and the swelling progressed to the thigh. Her ED-LOS was 21 h, and she received 10 vials of FHAV. Her complete remission occurred on the 25th day after the bite. (**A**) A POCUS study was conducted using a 15 MHz linear probe over the dorsal foot. This study revealed a clear borderline (dotted line) between the cobblestone sign (asterisks) and the normal tissue. (**B**) A series marker (arrowhead) was placed on the patient’s skin to indicate the proximal swelling margin. (**C**) The rate of proximal progression (RPP) was calculated by dividing the distance between the two markers by the number of hours. The RPP was measured in cm/hour.

**Figure 3 tropicalmed-08-00246-f003:**
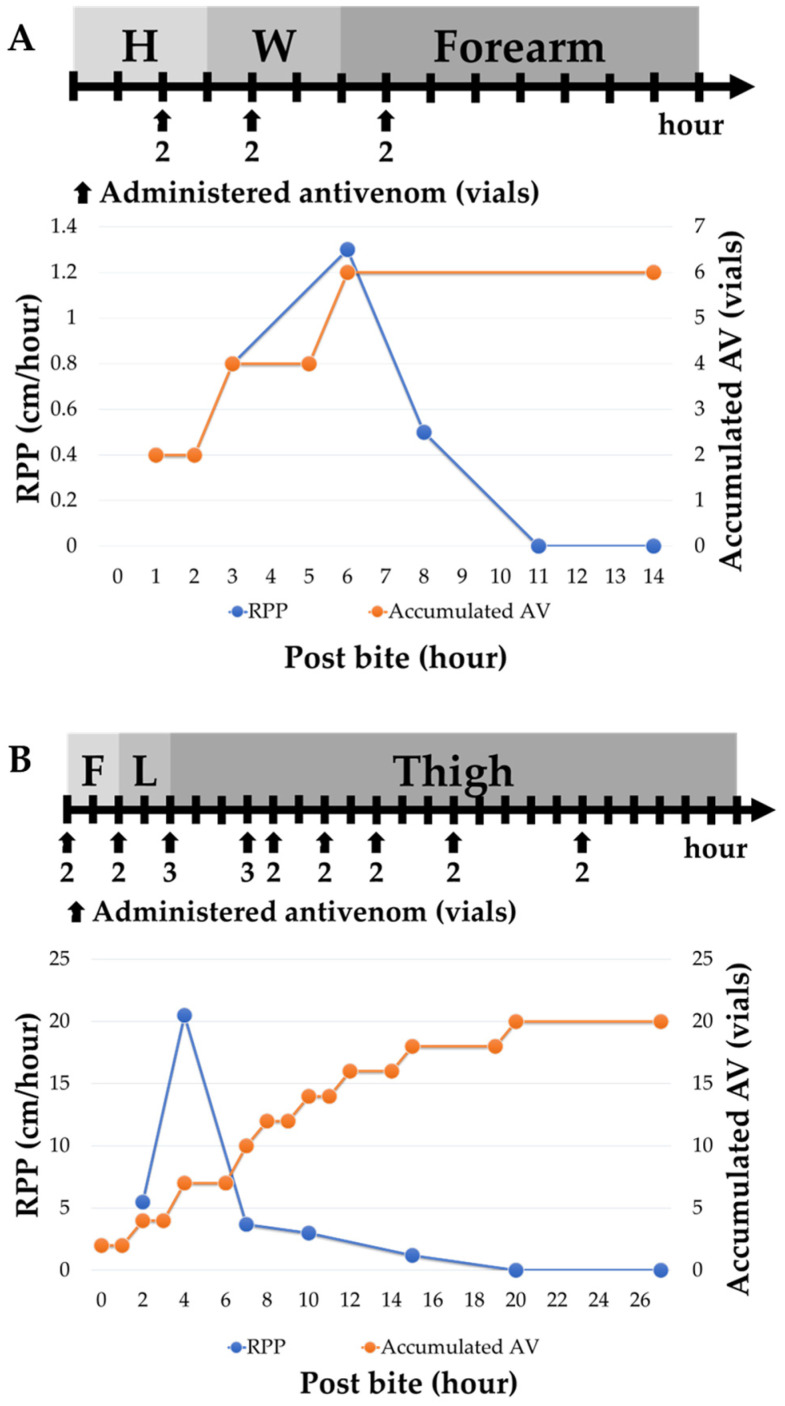
An upper bar chart summarizes the progression of edema by anatomical site in the hours after the bite. A line chart below shows the relationship between RPP (cm/hour) and the accumulated volumes of antivenom (vials). (**A**) An 81-year-old male belonging to Group IIA in Blaylock’s categories (considered mild) received a total of 6 vials of FHAV and an ED-LOS of 13 h. The patient achieved complete remission 14 days after treatment. (**B**) A 61-year-old male in Group IIB, classified as moderate according to Blaylock’s categories, received additional antivenom even though the RPP decelerated. He received a total of 20 vials of FHAV and had an ED-LOS of 27 h. Complete remission occurred 46 days after treatment. Abbreviations: F: foot; H: Hand; L: lower leg; W: Wrist.

**Figure 4 tropicalmed-08-00246-f004:**
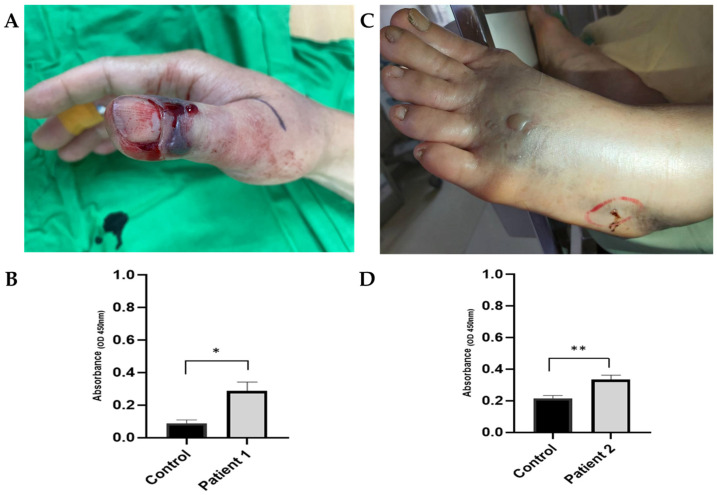
Patient 1, a 55-year-old male in Group IIA, was bitten on his right thumb and received 6 vials of FHAV. One day after the bite, he developed hemorrhagic bullae (**A**). The snake was identified as *P. mucrosquamatus* through venom analysis of the bullae fluid using purified anti-*P. mucrosquamatus* polyclonal IgY from egg yolk. The control was a BSA (bovine serum albumin) sample (**B**). Patient 2, a 71-year-old female in Group IIB, was bitten on her left foot and received 12 vials of FHAV. Blisters developed 11 h post-bite (**C**). The snake was also identified as *P. mucrosquamatus* through venom analysis of the bullae fluid (**D**). The error bar represents the standard deviation of the fraction. *, *p* < 0.05; **, *p* < 0.01.

**Table 1 tropicalmed-08-00246-t001:** Characteristic data, clinical variables, and laboratory findings in *P. mucrosquamatus*-bitten patients.

	Group I	Group II		Group II		
			*p*-Value	IIA	IIB	*p*-Value
Patients [persons, percentage (%)]	7 (24)	22 (76)		11 (50)	11 (50)	
Severe swelling condition (Blaylock category)			<0.0001			0.59
Minimal	7 (100)	0		0	0	
Mild	0	17 (77)		9 (81)	8 (73)	
Moderate	0	4 (18)		2 (19)	2 (18)	
Severe	0	1 (5)		0	1 (9)	
Gross	0	0		0	0	
Gender			0.06			0.08
Male	6 (86)	10 (45)		3 (27)	7 (64)	
Age (year-old)			0.65			0.21
Median	59.00	57.50		60.00	55.00	
Range	25–65	5–82		30–82	5–76	
Identify methods			0.54			0.67
Snake body	2 (29)	7 (32)		4 (36)	3 (27)	
Photo picture	5 (71)	12 (55)		5 (45)	7 (64)	
Toxin analysis	0	3 (14)		2 (18)	1 (9)	
Encountered season			0.63			0.12
Winter (12–2)	0	2 (9)		1 (9)	1 (9)	
Spring (3–5)	2 (29)	3 (14)		0	3 (27)	
Summer (6–8)	1 (14)	6 (27)		5 (45)	1 (9)	
Fall (9–11)	4 (57)	11 (50)		5 (45)	6 (55)	
Bite site			0.44			0.34
finger	0	2 (9)		1 (9)	1 (9)	
hand	4 (57)	3 (14)		1 (9)	2 (18)	
forearm	0	1 (5)		0	1 (9)	
toe	1 (14)	4 (18)		2 (18)	2 (18)	
foot	2 (29)	10 (45)		5 (45)	5 (45)	
lower leg	0	2 (9)		2 (18)	0	
Clinical presentation			0.06			1
Fever	0	0		0	0	
Local cyanosis	7 (100)	22 (100)		11 (100)	11 (100)	
Progressed swelling	0	22 (100)		11 (100)	11 (100)	
Wound necrosis	0	2 (9)		1 (9)	1 (9)	
Bullae/ Blister	0	2 (9)		1 (9)	1 (9)	
Laboratory results						
WBC [normal range: 4500–11,000 cells/μL]			0.12			0.06
WBC (>11,000)	0	6 (27)		5 (45)	1 (9)	
WBC (<11,000)	7 (100)	16 (73)		6 (55)	10 (91)	
PLT [normal range: 150,000–400,000 cells/μL]			0.41			0.14
PLT (<150,000)	0	2 (9)		2 (18)	0	
PLT (>150,000)	7 (100)	20 (81)		9 (82)	11 (100)	
CK [normal range: 39–308 U/L]			0.44			0.73
CK (>1000)	0	2 (9%)		1 (14)	1 (9)	
CK (<1000)	5 (71)	16 (73)		6 (86)	10 (91)	
Kindo grade renal function (eGFR, mL/min/1.73 m^2^)			0.38			0.17
Grade 1 (>90)	4 (57)	10 (45)		3 (27)	7 (64)	
Grade 2 (60–89)	3 (43)	7 (32)		4 (36)	3 (27)	
Grade 3 (30–59)	0	5 (23)		4 (36)	1 (9)	
Grade 4 (15–29)	0	0)		0	0	
Grade 5 (<15)	0	0		0	0	
Administered antivenom (vials)			<0.0001			<0.001
Median	2.00	9.50		6.00	12.00	
Range	1–4	4–22		4–10	8–22	
ED-LOS (hours)			<0.0001			0.03
Median	8.00	24.50		21.00	27.50	
Range	3–18	13–61		13–36	15.5–61	
Disposition			0.22			0.33
ED discharge	7 (100)	18 (82)		8 (73)	10 (91)	
admission and surgery	0	2 (9)		1 (9)	1 (9)	
admission without surgery	0	2 (9)		2 (18)	0	
Complete remission days			<0.001			0.06
Median	2.00	10.00		7.00	12.00	
Range	1–2	1–49		1–25	6–49	

Abbreviations: CK: creatine kinase; ED: emergency department; ED-LOS: emergency department length of stay; eGFR: estimated glomerular filtration rate; PLT: platelet; WBC: white blood cell.

## Data Availability

In cases where data is unavailable due to privacy or ethical restrictions.

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
