# Peer review of "Application of Sonographic Assessments of the Rate of Proximal Progression to Monitor Protobothrops mucrosquamatus Bite-Related Local Envenomation: A Prospective Observational Study"

_tropicalmed, 2023, doi:10.3390/tropicalmed8050246_

Round 1

Reviewer 1 Report

Main concern

1.     This is a study with innovation and deserved publication.

2.     However, methodological flaws need to be corrected before its acceptance for publication.

Material and methods

How Group II A and IIB were decided? Randomized ? or by chance? Please describe the method of grouping.

Statistics

“mean ± standard deviation (SD)” is unsuitable in this study.

Kruskal‒Wallis test is used for the median, not the mean. Therefore, please have new analysis results by using the correct statistical methods.  

Results

No RPP compared in your study group in Table 2. Please add it.

Group II comprised wide severity variations (from mild to severe). Can you have a further sub-group analysis of the mild case of RPP in Groups IIA and B? Because they are the main component of your study group? This analysis may clarify the current antivenom use in treating Taiwan habu bitten patients.

Table 1 is unsuitable for this study; describe Table 1 in the text. In addition, epidemiology data is unnecessary for this study; please delete it.

3.     Another review will be needed after the authors make changes to their manuscript.

4.     Other minor concerns are listed below.

Abstract

Line 28, “(FHAV) for alleviating local tissue injury remains unknown,” is not true.

Please read your reference 25 and correct it. https://doi.org/10.3390/toxins14110794

Introduction

Line 55 Local cyanosis is rarely seen in Taiwan habu snakebites patients. Also, the definition of “mild coagulopathy” ? “mild coagulopathy” is not easy to be seen in Taiwan habu patients. Therefore, please check them and correct them.

Author Response

Comments and Suggestions for Authors

Main concern

  1. This is a study with innovation and deserved publication.
  2. However, methodological flaws need to be corrected before its acceptance for publication.

Material and methods

How Group II A and IIB were decided? Randomized ? or by chance? Please describe the method of grouping.

Response: During the specified time, clinicians on duty had the discretion to administer the FHAV guided by the RPP, or not. Afterward, we retrospectively grouped the patients. Those who received treatment according to RPP guidelines were placed in group IIA, while those who were given more antivenom in an effort to reduce local tissue injury were placed in group IIB. For more information, please refer to Section 2.1, lines 106-109, and Section 5, Limitations, lines 388-390.

Statistics

“mean ± standard deviation (SD)” is unsuitable in this study.

Kruskal‒Wallis test is used for the median, not the mean. Therefore, please have new analysis results by using the correct statistical methods.  

Response: We greatly appreciate your valuable advice. We have corrected our statistical method and presented the continuous data in the form of median and range. For continuous factors, we used the Mann-Whitney U test and the Kruskal-Wallis test. For categorical variables, we used the chi-squared test. For more information, please refer to the following sections and lines: Abstract, lines 33-42; Section 2.4, lines 137-152; Section 3.2, lines 191-193, 212-222, and Table 1; Section 3.3, lines 247-249, 254-261.

Results

No RPP compared in your study group in Table 2. Please add it.

Response: Thank you for your valuable feedback. We did not include the RPP values in Table 1 because each patient had multiple series of RPP values, which would have made the table difficult to read. Instead, we selected one representative patient from each group (IIA and IIB) and presented their RPP values in Figure 3 for clarity. Please refer to Figure 3 for further information.

Group II comprised wide severity variations (from mild to severe). Can you have a further sub-group analysis of the mild case of RPP in Groups IIA and B? Because they are the main component of your study group? This analysis may clarify the current antivenom use in treating Taiwan habu bitten patients.

Response: Thank you for providing valuable feedback. We conducted a further analysis on the subgroup whose Blaylock's categories were classified as mild in group IIA and IIB. In this sub-analysis, group IIA received a lower volume of FHAV than group IIB (a median of 6 vials vs. 11 vials, with a p-value of <0.05). There were no significant differences found between genders (78% male vs. 50% male), ED-LOS (a median of 22 hours vs. 26.5 hours), or complete remission days (a median of 6 days vs. 12 days). The sub-analysis results have been included in section 3.3., lines 255-261.

Table 1 is unsuitable for this study; describe Table 1 in the text. In addition, epidemiology data is unnecessary for this study; please delete it.

Response: Thank you for your valuable input. We have removed Table 1 and incorporated the original findings into the text. Please see section 3.1 for further details.

  1. Another review will be needed after the authors make changes to their manuscript.
  2. Other minor concerns are listed below.

Abstract

Line 28, “(FHAV) for alleviating local tissue injury remains unknown,” is not true.

Please read your reference 25 and correct it. https://doi.org/10.3390/toxins14110794

Response:Thank you for your valuable input. Lin et al. demonstrated the potential of using FHAV to reduce swelling in affected limbs. However, in our study, we found that patients bitten by P. mucrosquamatus and receiving higher volumes of FHAV in the early stages did not necessarily experience shorter complete remission days. The timing of complete remission of local tissue injury was instead associated with the severity of local tissue swelling (Group I vs II, median complete remission days 2.00 days vs. 10.00 days, p value < 0.001). The volume of antivenom did not have a significant impact on complete remission days (Group IIA vs. IIb, median complete remission days 6 days vs. 12 days, p value > 0.05). Kindly refer to the results (line 220-222, 254-255) and discussion (lines 335-341) for more information. However, we acknowledge that, based on Lin et al.'s study, the appropriate dosage and efficacy of FHAV for alleviating local tissue injury remain uncertain. Refer to the abstract (line 28) for further discourse.

Introduction

Line 55 Local cyanosis is rarely seen in Taiwan habu snakebites patients. Also, the definition of “mild coagulopathy” ? “mild coagulopathy” is not easy to be seen in Taiwan habu patients. Therefore, please check them and correct them.

Response: Thank you for your kind advice. We have checked and corrected the paragraph. Please refer to lines 56-58.

Submission Date

07 March 2023

Date of this review

05 Apr 2023 11:27:55

Reviewer 2 Report

The manuscript "Application of sonographic assessment. of the rate of proximal progression to monitor Protobohrops mucrosquamatus bite-related local envenomation: a prospective observational study" is a well-written work which findings suggests to clinicians utilize RPP as an objective parameter to evaluate the timing of antivenom administration. Considering that snake envenoming is an important tropical and subtropical disease, which affects people of real communities in a high burden of mortality and morbidity, this research can contribute a lot to help clinicians to prescribe antivenin during snake bite envenoming. I have just one suggestion: review the legends of table 2, there are abbreviations that are not inside the table.

The main question addressed is the evaluation of the application of sonographic assessment to measure the progression of Protobothrops mucrosquamatus snake bite local envenomation. It is a huge field that any contribution is really wellcome. It is important to consider that each snake bite accident is specific for the specific species specimen, including several factors that can result in a specific local and systemic symptoms. So, this work can contribute to supplying information about local edema of more than 5 years of prospective observation of P. mucrosquamatus snake bite accidents. The results of this work  suggests that clinicians utilize RPP as an objective parameter to evaluate the timing of antivenom administration.   Besides the  characteristic data and clinical variables in snakebite patients analysis, the authors also collected laboratory findings in P. mucrosquamatus bit patients, which resulted in the suggestion of different clinical management for these ones based on the RPP.  Please, check the table 2 legends.

Author Response

I have just one suggestion: review the legends of table 2, there are abbreviations that are not inside the table.

Response: Thank you for acknowledging this article. We have added the abbreviation for Table 1 (the previous Table 2).

Reviewer 3 Report

Snakebite is a real challenge in many tropical countries. The current therapy  relies on only one scientific and effective approach: antivenoms. This approach has relevant limitations with serious implications for the clinical outcomes. Local pathological events are usually poorly neutralised by animal-derived antivenoms. The present investigation is novel and meets the ambitious plan of World Health Organization to tackle the huge impact of snakebites. 

1. The introduction should be restructured in a logical order in at least 2 paragraphs. In the current version, all information with different ideas is presented in only one paragraph. 

2. Quality of figures 2C, 3 and 4 should be improved. 

3. "More antivenom causes higher medical costs and possibly a higher incidence of serum sickness." Reference 20 does not support this sentence, especially the first half. 

4. I did not understand the following sentences: We hypothesized that FHAV may neutralize the lethal compartment of venom but not be effective for  the compartment leading to the progression of local tissue injury. The affinity of FHAV to the different compartments of P. mucrosquamatus still needs further evaluation. The antivenom neutralizing tear about which compartment of venom and local cytotoxic assay still need further evaluation. Compartment of venom?

5. There are many factors behind the poor efficacy of antivenoms against some local pathological events. Authors should expand the discussion around this topic, taking account some previous preclinical evaluation of antivenoms. 

6. Line 277: Scientific names are always italicised.

7. The limitations of the study should be included and discussed.

8. The results and discussion are not focused on the main suggestion or finding of the present investigation. 

Author Response

Comments and Suggestions for Authors

Snakebite is a real challenge in many tropical countries. The current therapy  relies on only one scientific and effective approach: antivenoms. This approach has relevant limitations with serious implications for the clinical outcomes. Local pathological events are usually poorly neutralised by animal-derived antivenoms. The present investigation is novel and meets the ambitious plan of World Health Organization to tackle the huge impact of snakebites. 

  1. The introduction should be restructured in a logical order in at least 2 paragraphs. In the current version, all information with different ideas is presented in only one paragraph. 

Response: Thank you for your valuable advice. We have divided the introduction into three paragraphs in a logical order. Please see the updated introduction below.

  1. Quality of figures 2C, 3 and 4 should be improved. 

Response: We had improved the quality of figures.

  1. "More antivenom causes higher medical costs and possibly a higher incidence of serum sickness." Reference 20 does not support this sentence, especially the first half. 

Response: Thank you for bringing our mistake to our attention. We used the wrong reference initially, but we have corrected it. Please refer to section 4, lines 324 and reference 20 and 21.

  1. I did not understand the following sentences: We hypothesized that FHAV may neutralize the lethal compartment of venom but not be effective for  the compartment leading to the progression of local tissue injury. The affinity of FHAV to the different compartments of P. mucrosquamatus still needs further evaluation. The antivenom neutralizing tear about which compartment of venom and local cytotoxic assay still need further evaluation. Compartment of venom?

Response:Thank you for your kind feedback. The toxin profiles of P. mucrosquamatus include snake venom metalloproteinase (SVMP; 29.4%), C-type lectin (CLEC; 21.1%), snake venom serine protease (SVSP; 17.6%), and phospholipase A2 (PLA2; 15.9%). We hypothesize that FHAV may be effective in neutralizing the lethal component of the toxin, possibly SVMP, but not the component that leads to local tissue injury, possibly CLEC. This phenomenon was also observed in the neutralization between FNAV and the toxin of Naja atra. FNAV had high affinity for NTX, but CTX played a major role in dermonecrosis. We apologize for using the wrong word "compartment." We've corrected it to "component." Please see section 4, lines 366-372 for the updated information.

  1. There are many factors behind the poor efficacy of antivenoms against some local pathological events. Authors should expand the discussion around this topic, taking account some previous preclinical evaluation of antivenoms. 

Response: Thank you for your advice. We have rewritten the discussion on the efficiency of antivenom in neutralizing snakebite-related local injuries. Please refer to the discussion on lines 338-381.

  1. Line 277: Scientific names are always italicised.

Response: We have corrected the error. Please refer to line 303.

  1. The limitations of the study should be included and discussed.

Response: Thank you for your valuable feedback. We have already included a section on the limitations of our study. Please refer to Section 5: Limitations, lines 389-398.

  1. The results and discussion are not focused on the main suggestion or finding of the present investigation. 

Response: Thank you for your helpful advice. We have revised and strengthened our suggestions in the discussion and conclusion. Please refer to the abstract (lines 42-45), discussion (lines 317-321), and conclusion sections for the updated information.

Submission Date

07 March 2023

Date of this review

08 Apr 2023 13:06:04

Round 2

Reviewer 1 Report

Do you really use “a median of 9.50 vials ” I mean, antivenom almost be administrated in 1, 2, 3, 4,….vials but not 1.2, 2.5,…. Therefore, please correct it.

Author Response

Comments and Suggestions for Authors

Do you really use “a median of 9.50 vials ” I mean, antivenom almost be administrated in 1, 2, 3, 4,….vials but not 1.2, 2.5,…. Therefore, please correct it.

Response: Thank you for your helpful suggestion. We agree that the statement about the volume of antivenom may have caused confusion among readers. To clarify this issue, we have added a more detailed explanation about the administration of antivenom. Please see section 2.2, lines 116-118 for further information.

Reviewer 3 Report

I consider this new version of the manuscript acceptable for publication. 

Author Response

Comments and Suggestions for Authors

I consider this new version of the manuscript acceptable for publication. 

Response: Thank you for appreciating our work. Thank you for your valuable opinions during the article revision.
